# Characterization of Myeloperoxidase in the Healthy Equine Endometrium

**DOI:** 10.3390/ani13030375

**Published:** 2023-01-21

**Authors:** Sonia Parrilla Hernández, Thierry Franck, Carine Munaut, Émilie Feyereisen, Joëlle Piret, Frédéric Farnir, Fabrice Reigner, Philippe Barrière, Stéfan Deleuze

**Affiliations:** 1Physiology of Reproduction, Faculty of Veterinary Medicine, University of Liège, 4000 Liège, Belgium; 2Center for Oxygen Research and development (CORD), University of Liège, 4000 Liège, Belgium; 3Laboratory of Tumor and Developmental Biology, GIGA-Cancer, University of Liège, 4000 Liège, Belgium; 4Department of Morphology and Pathology, Faculty of Veterinary Medicine, University of Liège, 4000 Liège, Belgium; 5Biostatistics and Bioinformatics Applied to Veterinary Sciences, Faculty of Veterinary Medicine, University of Liège, 4000 Liège, Belgium; 6UE1297 PAO, INRAE, 37380 Nouzilly, France; 7Equine and Companion Animal Reproduction, Veterinary Medicine Faculty, University of Liège, 4000 Liège, Belgium

**Keywords:** myeloperoxidase, equine, immunohistochemistry, endometrial mucosal immune system

## Abstract

**Simple Summary:**

Endometritis or endometrial inflammation is a major cause of infertility in mares. Understanding how the immune system protects the uterus against potential pathogens and contamination is essential for preventing endometritis and other associated pathologies. Myeloperoxidase (MPO) is an enzyme mainly contained in inflammatory cells and indeed associated with equine endometritis. Surprisingly, this enzyme is also detected in the uterus of mares in absence of inflammation. The aim of this study was to investigate MPO in the uterus of mares in physiological conditions throughout the reproductive cycle to better understand its function in equine reproduction. MPO is constantly present in the uterus of mares in all phases of the reproductive cycle. This MPO, in absence of inflammatory cells, seems to be synthesized and secreted in the uterine lumen by the uterine cells themselves, especially in estrus, when the equine uterus is most exposed to external contamination. Based on these results and its potent bactericidal action, we suggest that MPO is probably part of the uterine immune system protecting the uterus against contamination and avoiding inflammation.

**Abstract:**

Myeloperoxidase (MPO), as a marker of neutrophil activation, has been associated with equine endometritis. However, in absence of inflammation, MPO is constantly detected in the uterine lumen of estrous mares. The aim of this study was to characterize MPO in the uterus of mares under physiological conditions as a first step to better understand the role of this enzyme in equine reproduction. Total and active MPO concentrations were determined, by ELISA and SIEFED assay, respectively, in low-volume lavages from mares in estrus (*n* = 26), diestrus (*n* = 18) and anestrus (*n* = 8) in absence of endometritis. Immunohistochemical analysis was performed on 21 endometrial biopsies randomly selected: estrus (*n* = 11), diestrus (*n* = 6) and anestrus (*n* = 4). MPO, although mostly enzymatically inactive, was present in highly variable concentrations in uterine lavages in all studied phases, with elevated concentrations in estrus and anestrus, while in diestrus, concentrations were much lower. Intracytoplasmic immunoexpression of MPO was detected in the endometrial epithelial cells, neutrophils and glandular secretions. Maximal expression was observed during estrus in mid and basal glands with a predominant intracytoplasmic apical reinforcement. In diestrus, immunopositive glands were sporadic. In anestrus, only the luminal epithelium showed residual MPO immunostaining. These results confirm a constant presence of MPO in the uterine lumen of mares in absence of inflammation, probably as part of the uterine mucosal immune system, and suggest that endometrial cells are a source of uterine MPO under physiological cyclic conditions.

## 1. Introduction

The pro-oxidant enzyme Myeloperoxidase (MPO) is one of the key mediators in neutrophil functions [1] and considered as a marker of neutrophil activation. It contributes to neutrophil defense functions by the production, in presence of H_2_O_2_, of potent oxidant agents responsible for microbial cell damage [2]. MPO is mainly contained in azurophilic granules of neutrophils [3] playing an important role in the intracellular microbial killing after phagocytosis. It is also involved in the extracellular clearance of several pathogens when released into the extracellular medium by degranulation, cell lysis as well as a constituent of neutrophil extracellular traps (NETs) [4]. Nevertheless, despite their protective role as a potent antimicrobial, when overabundant or upregulated in the extracellular medium, the oxidant capacities of MPO products can be detrimental for host cells and tissues [2]. Accordingly, MPO has been associated with the development of some inflammatory and fibrotic diseases in different organs [5,6,7,8]. The synthesis and expression of MPO have been exclusively related to myeloid cells, mainly neutrophils [9]. However, MPO has recently been described to be contained in non-myeloid cells [10,11,12,13]. Non-inflammatory cells are able to capture MPO from the extracellular medium [14,15], and through direct contact with neutrophils [16]. More interestingly, the capacity of non-myeloid cells to express endogenous MPO has been demonstrated in cells of the reproductive tract of rams and human neurons under physiological conditions [10,11], as well as in human endothelial cells as a response to oxidative stress [17].

In horses, MPO has been identified, both under active and inactive form, in various fluids and tissues [18,19,20,21] and has been shown to be involved in numerous inflammatory processes [18,19,22,23,24]. Some studies have focused on MPO and its implications in pathological conditions of the endometrium, such as endometritis and endometrosis [24,25,26,27], which are a major cause of infertility in the mare that adversely impact the horse breeding industry. Parrilla-Hernandez et al. [24] showed that in the equine endometrium in estrus, MPO concentrations were higher when associated with a positive endometrial cytology. However, in the same study, MPO was also detected at variable concentrations in all the studied mares even in the absence of neutrophils. Similarly, an association between MPO and the number of neutrophils has not been observed in uterine samples of cows for several weeks after calving [28]. These results suggest a constitutive presence of MPO, not exclusively related to inflammation [24] that has yet to be investigated. While presence of MPO in the endometrium has been observed by immunohistochemistry and Western blot in rats, before and after an induced inflammation [13], no such report has been published in mares.

MPO has also been studied as a component of NETs in the equine endometrium and it has been suggested to be involved in the development of endometrial fibrosis [25,26,29]. Thus, the presence of MPO in the uterus of mares raised the question about a possible pathological action of the enzyme in the equine endometrium and its association with different equine endometrial diseases. The measurement of MPO in uterine fluid as a protein provides information about the abundance of the molecule, under its native form or its precursor. However, the enzymatic activity, defined as the capacity of the enzyme to produce oxidant agents, can vary considerably between individuals [30,31] and is directly responsible for the effects on the equine endometrium. This has been demonstrated in vitro, in a study that showed that the inhibition of the enzyme decreases the MPO pro-fibrotic effect [29]. Therefore, the evaluation not only of the enzyme but also of its activity is crucial to understand its possible effect in mares’ endometrium.

This study aims to characterize the presence, activity and localization of MPO in the healthy uterus of mares during the different phases of the reproductive cycle, as a first step to understand its physiological and/or pathological role in equine reproduction.

## 2. Materials and Methods

All procedures on animals were conducted in accordance with the guidelines for the care and use of laboratory animals issued by the French Ministry of Agriculture and with the approval of the ethical review committee (Comité d’Ethique en Expérimentation Animale Val de Loire) under number APAFIS#5084-201604142152987 v2.

### 2.1. Animals and Samples Collection

The study was performed on adult research pony mares housed in the Experimental Unit of Animals Physiology of l’Orfrasière (UEPAO; INRAE Val de Loire 1297, Nouzilly, France) between March 2016 and January 2017. Thirty-six Welsh pony mares, aged 5 to 18 years were included and 57 sets of samples were recovered. At sampling, seven mares were maiden while the other 29 mares had foaled at least once. Samples were obtained from mares once (*n* = 18), twice (*n* = 15) or three times (*n* = 3) during the reproductive season.

Mares were examined by transrectal palpation and ultrasonography for genital health and to determine the reproductive stage. Data and samples were obtained during the breeding season from 29 mares in estrus (showing a dominant follicle and endometrial oedema) and 18 mares in diestrus (presenting a corpus luteum in absence of oedema). Ten mares in seasonal anestrus (exhibiting no follicle greater than 15 mm and no corpus luteum) were also included in the study. Blood samples were collected by jugular venipuncture from those mares and plasma concentration of progesterone lower than 1 ng/mL [32] on two evaluations seven days apart confirmed the seasonal anestrus.

After scrubbing of the vulva and peritoneum, a low-volume uterine lavage was performed with 60 mL of saline solution using a bovine embryo flushing catheter (Minitüb, Tiefenbach, Germany). After a transrectal uterine massage, the fluid was collected in a 50 mL graduated conical tube by gravity flow and samples were centrifuged at 600 g for 10 min immediately. Supernatants were collected in 2 mL tubes and kept at –20 °C for further analysis.

Immediately after the lavage, an endometrial biopsy sample was obtained from the proximal region of one uterine horn using an alligator jaw biopsy instrument (Krusse, Langeskov, Denmark). Biopsy samples were fixed in 10% formalin and then embedded in paraffin.

### 2.2. Histology

Endometrial biopsies were stained with hematoxylin and eosin for histopathological examination by light microscopy. Slides were examined for the presence of neutrophils within the luminal epithelium and the stratum compactum. Infiltration of three or more neutrophils in estrus [33] and one or more neutrophils in diestrus [34,35] and anestrus [34] per five fields at high magnification (40×) was considered as evidence of endometritis. Additionally, biopsy specimen were routinely classified using the grading system of Kenney and Doig [36], modify by Schoon et al. [37].

### 2.3. Total Myeloperoxidase by ELISA Assay

Concentrations of total MPO in the supernatant of the low-volume uterine lavages were determined by a commercial equine ELISA kit (Bioptis SA, Liege, Belgium) as previously described [24,38]. After a preliminary screening, each sample was assayed at the dilution 50× and/or 500×. Each sample was assayed twice and the mean value was calculated. MPO concentration was below the detection limit (<1.3 ng/mL) for 3.8% of the samples. Concerning the remaining 96.2% of samples with a detected MPO concentration, 68% had an intra-assay coefficient of variation (CV) between 0.6% and 18.7%. Samples with higher CVs mainly correspond to samples assayed at both dilutions. For those samples, the concentration corresponding to the dilution 50× was considered.

### 2.4. Active Myeloperoxidase by SIEFED Assay

Active MPO concentration in the supernatant of the low-volume uterine lavage was determined by specific immuno-extraction followed by enzymatic detection (SIEFED) assay [39]. Each sample was assayed twice and the mean value was calculated. Active MPO was below the detection limit (<0.051 ng/mL), in 23% of the samples. For the remaining samples (77%), 70% of them showed active MPO with intra-assay CVs ranged between 0.6 and 15.4%. The samples showing CVs greater than 16% mainly correspond to those situated at the lower limit of detection of the calibration curve.

### 2.5. Protein Concentration

The total protein concentration in the supernatant of low-volume uterine lavage was measured by the bicinchoninic acid method for protein determination (Sigma-Aldrich, Inc., St. Louis, MO, USA).

### 2.6. Immunochemistry

Four micrometer thick tissue sections of paraffin blocks from mares in estrus (*n* = 11), diestrus (*n* = 6) and anestrus (*n* = 4) were deparaffinized and rehydrated following a classical protocol. Heat induced epitope retrieval was carried out in target retrieval solution pH 6 (Dako S1699, Santa Clara, CA, USA,) in autoclave for 11 min at 126 °C followed by a cool down for 20 min and rinse in distilled water for 5 min at room temperature. Endogenous peroxidases were subsequently blocked by 3% H_2_O_2_/H_2_O for 20 min at room temperature and slides were then rinsed with distilled water (2×). To inhibit non-specific staining, samples were incubated in Protein Block Serum Free (Dako X0909, Santa Clara, CA, USA) at room temperature for 10 min. Slides were first incubated for 60 min at room temperature with rabbit anti-MPO antibody obtained against purified equine MPO [38] (1/1500) and then with a HRP-conjugated goat anti-rabbit secondary antibody (Envision System-Labeled Polymer-HRP,DAKO, K4003), for 30 min at room temperature. After washing with PBS 5 times for 5 min, color development was achieved by applying diaminobenzidine tetrahydrochloride (DAKO K3468, Santa Clara, CA, USA) solution for 3 min. Slides were finally counterstained with hematoxylin and protected by a coverslip for microscope observation. Slides from a confirmed case of equine meningitis were used as a positive control. For negative controls, the primary antibody was omitted.

### 2.7. Semiquantitative Evaluation of the Myeloperoxidase Immunoexpression

For assessment of a precise MPO immunostaining in endometrial tissues, the percentage of immunopositive cells as well as their staining intensities were used to determine the immunoreactive score (IRS) (adapted from [40]). To this purpose, the percentage of immunopositive cells (PP) within five representative areas (40×) of each cell population was determined. For each immunostained cell, the staining intensity (SI) was evaluated and assigned a numerical value (0 = none; 0.5 = very weak or dot-like pattern; 1 = weak; 2 = moderate; 3 = strong). The IRS was determined according to the formula:5
IRS = 1/100∑{PPn × SIn}
*n* = 1

This was performed in general for each endometrial tissue sample and separately for the following epithelial cell populations: the luminal epithelium (LE), the glandular ducts (GD), as well as the mid glands (MG) and basal glands (BG).

### 2.8. Statistical Analysis

In order to limit the effect of dilution and allow comparison between samples, results are expressed as total or active MPO/total protein ratio, referred to as R_T_ and R_A_, respectively. A general linear mixed model (SAS version 9.3) was used to compare data between groups including post hoc test and results are expressed as least square means. When the model did not converge, a general linear model was used to compare groups. GraphPad Prism version 9 was used to perform other statistical analyses. Normal distribution of parameters was tested with the Shapiro–Wilk normality test. Values of MPO concentration and total protein were non-normally distributed, and IRS values were normally distributed. Correlations were evaluated using the Spearman test. Statistical significance was established at *p*-value less than 0.05.

## 3. Results

### 3.1. Histology Evaluation

A small number of smears showed tissular infiltration of neutrophils in endometrial biopsies (estrus *n* = 3; diestrus *n* = 0; anestrus *n* = 2). Since uterine biopsy is accepted as a “gold standard’’ for the diagnosis of endometritis, samples were considered physiological and included in the study when no inflammation was detected using this method. A total of 52 samples: eight samples obtained in anestrus, 18 in diestrus and 26 in estrus, were then included. All mares presented no or mild endometrial fibrosis except one showing a moderate degree of fibrosis.

### 3.2. Myeloperoxidase

Myeloperoxidase was detected in all uterine lavage fluids except for a sample obtained in diestrus and a sample obtained in anestrus whose concentration of total MPO was below the detection limits of the assay. Active MPO concentration was below the detection limits of the assay for 12 samples across all groups. Values of active and total MPO concentration are shown in Table 1. The estrus phase showed higher concentrations of total MPO compared to the other groups. Maximal values of active MPO were observed in anestrus.

### 3.3. Protein Concentration

The total protein concentration in uterine lavages (Table 1) varied individually, but no statistical differences were observed between groups.

Concentration of total MPO was positively correlated with the total protein concentration in estrus (*r* = 0.6623; *p* < 0.001) and in diestrus (*r* = 0.7155; *p* < 0.001), but not in anestrus. However, no correlation between total protein and active MPO concentrations was observed in any of the groups (Figure 1).

### 3.4. Total Myeloperoxidase/Total Protein Ratio (R_T_)

R_T_ for the different groups is given in Table 1. R_T_ values varied between samples and groups, with values ranging from 243.57 to 54,807 ng/mg of proteins. A significant difference between the studied groups (*p* < 0.005) was observed. In diestrus, R_T_ was significantly lower than in estrus (*p* < 0.005) and anestrus (*p* < 0.005). However, no significant differences were found between the anestrus and estrus groups (Figure 2a).

### 3.5. Active Myeloperoxidase/Total Protein Ratio (R_A_)

R_A_ values for the different groups are given in Table 1. When active MPO was detected, R_A_ was generally low, with values ranging from 0.008 to 72.21 ng/mg of proteins.

We observed a significant difference of R_A_ between the studied groups (*p* < 0.0001). R_A_ was higher in anestrus than in estrus (*p* < 0.0001) and diestrus (*p* < 0.0001). Nevertheless, no significant differences were found between the estrus and diestrus groups (Figure 2b). No correlation was observed between R_T_ and R_A_ for any of the groups.

### 3.6. Specific Activity of Myeloperoxidase

Specific activity of MPO was expressed as the ratio of active MPO versus total MPO measurements. Values were very low in all studied samples and showed a high within-group variability but no statistical differences between groups were observed (Figure 2c).

### 3.7. Immunohistochemistry

#### 3.7.1. MPO Expression in the Endometrium

Neutrophils were immunohistologically stained in all samples working as a positive control of the MPO immunostaining. MPO staining was also detected in epithelial and stromal cells in the endometrium of mares as well as in secretory products within the glandular lumen. Endometrial expression of MPO was different between mares and was influenced by the reproductive phase (Figure 3).

Maximal MPO immunoreaction was detected during estrus. In this phase of the cycle, a widespread expression of MPO was observed in uterine glands where the staining intensity increased with depth (Figure 4A). Middle to deep glands showed a uniform, predominantly apical, reaction pattern (Figure 4C), whereas in the duct parts of the glands, the cytoplasmic staining was mostly diffuse. A nuclear staining of individual cells could occasionally be detected in endometrial glands. In diestrus, cytoplasmic MPO expression was not generalized and only some glands (mostly basal) were immunopositive (Figure 4F,H). Glandular cells showing a simultaneous cytoplasmic and nuclear or solely nuclear MPO immunostaining were more frequently found in diestrus than in estrus (Figure 4I).

The adluminal epithelium was irregularly marked, although a more generalized pattern was observed in estrus when compared to diestrus. An intracytoplasmic diffuse mosaic-like staining reaction was observed in immunopositive areas where cells with different staining intensities were mixed with immunonegative cells (Figure 4B,G). Nevertheless, in some mares presenting neutrophils in sub and transepithelial areas, a uniform cellular immunostaining was predominant (Figure 4D).

Stromal cells, mostly in the stratum compactum, were only scarcely immunopositive, showing a diffuse pattern in both phases of the cycle (Figure 4B).

In anestrus, only residual perinuclear MPO staining could be observed in luminal epithelial cells while the rest of the endometrial cell populations were immunonegative (Figure 4E).

#### 3.7.2. Semiquantitative MPO Immunoexpression

General immunostaining of MPO, evaluated by the IRS, was statistically greater in estrus than in other groups. Similarly, except for luminal epithelium in diestrus, the IRS for each cell sub-population was statistically higher in estrus than during the other phases, with maximal differences found in mid and basal glands. No such differences were observed between diestrus and anestrus (Figure 5).

In estrus, the maximal MPO expression was observed in basal glands. This was not observed in diestrus where IRS values between basal glands and luminal epithelium were not statistically different (Figure 6). MPO expression in endometrial tissues did not correlate with either R_T_ or R_A_ in the uterine fluid for the different studied cell populations and groups.

## 4. Discussion

Myeloperoxidase is mainly contained and released by neutrophils and hence considered as a marker of inflammation in different organs and tissues [5]. Presence of MPO in uterine lavage fluids at variable concentrations has been demonstrated in mares during estrus regardless of the presence of neutrophils [24]. This is concordant with the results of the present study, where total MPO (R_T_) was detected at highly variable concentrations in uterine lavage fluids of mares in all phases in physiological conditions. This suggests a consistent presence of MPO in the equine uterine lumen, not exclusively related to inflammation.

The main source of MPO in blood are neutrophils [41]. However, the number of neutrophils in our samples were within the physiological limits in mares for every reproductive phase [33,34,35,42]. In a healthy, non-inflamed mare uterus, only few neutrophils may be found in the endometrium during estrus, while during diestrus, they are practically non-existent [42]. For a long time, it was accepted that MPO gene expression, and consequently its biosynthesis, was exclusively associated with myeloid precursors [9], but recently the enzyme has been localized in some non-myeloid cells in inflammatory and physiological conditions in different organs [10,11,12,13]. For the first time, our study shows the immunohistochemical expression of MPO in equine endometrial cells.

MPO was contained in epithelial, and to a lesser extent, stromal cells, showing different staining patterns depending on the cell populations, suggesting functional differences between the distinct endometrial cells, as observed for other proteins [43,44,45,46]. Mid and basal glands showed maximal MPO expression and an apical reinforcement of the staining that, together with the fact that intra-luminal glandular secretions were also immunopositive, suggest an intracellular synthesis followed by an intraluminal secretion of MPO by uterine glands [12,43,47]. Maximal MPO expression was detected in estrus all over the endometrium, while in diestrus, an important diminution of staining and sporadic distribution of the immunoreaction were observed as illustrated in Figure 3. This is concordant with the higher concentrations of MPO (R_T_) in uterine lavage fluids detected in estrus compared to diestrus. Although this should be confirmed by transcriptomic and proteomic approach, such as in situ hybridization analysis, these results suggest that the endometrium may be a source of MPO in absence of endometritis.

In contrast to middle and basal glands, the diffuse MPO immunoreaction observed in the adluminal epithelium suggests an internalization of the enzyme rather than an intracellular synthesis. The two different patterns of immunostaining observed in the adluminal epithelium suggest different ways of MPO uptake by the epithelial cells. In zones where sub or transepithelial neutrophils were observed, epithelial cells were consistently positively stained, which could represent an MPO uptake by epithelial cells from neutrophils via cell-to-cell contact, as demonstrated in vitro in endothelial cells [16]. However, epithelial cells from zones of the adluminal epithelium where neutrophils were absent displayed no staining or a mosaic pattern. These variable intensities of labeling, may either illustrate the various amounts of MPO remaining after such an uptake or, alternatively, may result from an internalization of intraluminal MPO, as demonstrated in vitro for endothelial and lung epithelial cells [14,15]. Accordingly, the stromal cells were immunopositive in some samples probably also due to an MPO uptake as endometrial stromal cells appear to remove extravasated material from endometrial interstitium [48].

Some cells, mainly located in basal glands in diestrus samples, seemed to show a nuclear immunostaining occasionally associated with cytoplasmic staining. The nuclear localization can indicate a nuclear translocation of MPO and its involvement in the regulation of gene transcription as proposed for β-defensins in the equine endometrium [45]. As this nuclear staining was mostly observed in diestrus samples, it can be proposed that MPO nuclear translocation acts as a self-regulation mechanism targeting its gene transcription. At this point, our results do not allow to determine whether MPO adheres to the nuclear membrane as shown in horse muscle cells coincubated with MPO [49] or actually reaches inside the nucleus as demonstrated on murine peritoneal B-lymphocytes [50]. Further studies are necessary to confirm the exact localization of MPO at the nuclear level and to understand the functional significance of this nuclear staining.

Epithelial cells of the endometrium of mares, in addition to form an uninterrupted mucosal barrier between the lumen and the underlying cells and tissues, produce a variety of substances that maintain endometrial homeostasis and participate in reproduction [51]. Among others, uterine epithelial cells produce a range of natural unspecific microbicides, such as lactoferrin [52] and lysozyme [53], that confer protection against potential pathogens to uterine secretions and mucus. Those substances play an important role during the long estrus period in mares, when the cervix is open and relaxed and the ascending contamination of the uterus is facilitated. Although further investigations are needed to confirm it, based on the elevated concentration, expression and the apical staining pattern we observe in estrus, it seems logical to propose that MPO, being a potent bactericidal protein, is synthetized and secreted by the equine endometrium to participate to the mucosal immune system. Whether the secretion of MPO in the uterine lumen is constitutive and/or induced by a stimulus still needs to be investigated. In a study of Yang et al. [13] the immunohistochemical expression of MPO decreased in the endometrium of rats when a LPS-inflammation was induced. It can be speculated that this decrease of MPO expression is a consequence of a release of the intracytoplasmic MPO into the uterine lumen in reaction to the LPS-induced inflammation. Together with the constitutive presence of MPO in the equine endometrium observed in our study, these results suggest that MPO may be expressed by the endometrium to prevent a possible uterine contamination and inflammation, and is secreted into the uterine lumen in response to specific situations. This could partly explain the discrepancy between the intensity of the immunostaining and the intraluminal MPO (R_T_) concentrations in our physiological samples, as illustrated by the lack of correlation between the IRS and MPO (R_T_) concentrations. Functional data studies are necessary to further investigate the role of MPO in the uterus of mares and the consequences of its dysfunction.

The cyclical expression of MPO, as well as the total absence of glandular staining in endometrial cells in anestrus, is concordant with a protein expression regulated by steroid hormones. The activity of MPO in blood neutrophils seems to be regulated by the menstrual cycle in women [54,55] and an upregulation of MPO activity by estrogens has been suggested, probably through the direct stimulation of the expression of the MPO gene [55]. In uterine tissues of rats, MPO activity was reported to increase during the follicular phase of the cycle, decrease thereafter during the luteal phase, and reach minimal values in late diestrus samples [56]. However, in this paper, authors analyzed the peroxidase activity, which is common to all peroxidases and not an exclusive reaction of MPO. Nevertheless, these results agree with different studies showing that peroxidases in uterine tissue are upregulated by estrogens [57,58,59,60] via their receptors [61], while progesterone inhibits the estrogen-induction of uterine peroxidases. Our results show that a similar pattern of regulation also applies to MPO in the equine endometrium as the immunolabeling was marked in estrus, decreased in diestrus, and totally faded away in anestrus. This is further supported by the positive correlation between the concentration of MPO and total protein we observe in estrus and diestrus when steroids are secreted, while no such correlation exists in anestrus in the absence of significant concentrations of steroids.

Endometrial neutrophil population during anestrus has not been studied specifically yet. When evaluating the number of neutrophils as a marker of inflammation, the same threshold is often used in anestrus and diestrus because the blood–uterine barrier permeability is considered basal during these two phases, while it is increased under estrogen’s influence [62]. However, this neglects the effect of progesterone in suppressing uterine immune defense [63], which may depress the migration of neutrophils into the uterine lumen [64]. In anestrus, when the uterus is not under the influence of steroid hormones, the immunity of the endometrium may be similar to other mucosal systems. In fact, our study showed that in anestrus samples, neutrophils were usually observed in subepithelial capillaries with sporadic transepithelial passage in a mare-dependent manner. This presence of neutrophils in the mucosal endometrium is probably physiological and adapted to the endometrial needs to control microbiota [65] as well as to face any possible contamination [63,66]. Activation of these endometrial neutrophils, as a part of the mucosal defense system, may be the origin of some of the MPO (R_T_) encountered in anestrus samples.

Intraluminal accumulation of MPO from successive cycles and/or previous endometritis may contribute to the intraluminal MPO and may explain individual differences in MPO (R_T_) concentrations observed between mares. The intraluminal fluid accumulation in mares during estrus has been associated with higher concentrations of MPO [24]. An impaired drainage of endometrial luminal fluid through the cervix and/or the lymphatic system may reduce the elimination of MPO from the uterus and favor its accumulation from one cycle to the next. Thus, the individual mare’s capacity to eliminate MPO from the uterine lumen may explain the higher concentrations of MPO (R_T_) in some diestrus samples, when for others, levels of the enzyme were below the limit of detection of the assay. Moreover, the accumulation of MPO originating from previous physiological cyclic phases or inflammatory episodes may explain the high concentrations of MPO (R_T_) observed in some anestrus mares.

MPO has an important protective role in the innate immune defense [67]. However, by the same processes used in the destruction of pathogens, extracellular MPO can be detrimental for host cells and tissues (recently reviewed in [68,69]). Therefore, the constant presence of MPO in the uterine lumen raised the question about its possible contribution to the development of some endometrial pathologies. It has been suggested that MPO could be involved in the development of fibrosis in the equine endometrium [25,26]. Endometrial fibrosis is somehow connected to endometrial inflammation as persistent post breeding endometritis, which results in enhanced neutrophil recruitment and leads to increased fibrosis in mares [70]. In fact, mares diagnosed with endometritis present higher concentrations of MPO than those where no inflammation was observed [24]. Nevertheless, maiden mares, whose uterus has not been exposed to semen during their entire reproductive life, also develop endometrial fibrosis and its severity is correlated with age [71]. Thus, we can speculate that the constitutive expression and secretion of MPO by epithelial endometrial cells during the reproductive life may contribute to the natural progress of endometrial fibrosis, while the exposure to inflammation, particularly if persistent, and the subsequent increased release of MPO may accelerate and/or aggravate this process.

Previously, some mares with no sign of endometritis have been shown to have high concentrations of total MPO in the uterine lumen, similar to those presenting inflammation [24]. In the present study, MPO in uterine lavage fluids was mostly enzymatically inactive. Looking at the presence of active MPO (R_A_) in the different reproductive phases, our results suggest the presence of regulating factors in the uterine fluid balancing the MPO activity to best suit the endometrial requirements for every reproductive situation and avoid the detrimental effects for host cells and tissues. A significant fraction of MPO is inactivated during phagocytosis when released in the extracellular medium [72,73] but the involved mechanisms of inhibition are not clear yet. In addition, some molecules seem to be able to inhibit plasma MPO by different mechanisms and at different degrees [74,75,76,77] but, to our knowledge, they have not been studied in uterine fluid yet. Furthermore, in equine sperm supernatant, where high concentrations of total MPO are also observed, Ponthier et al. [20] showed an important presence of MPO precursor. Unlike in the human [2], the enzyme precursor is inactive in the equine [38] and is then cleaved into the active subunit of the enzyme. Thus, the low MPO activity measured in the uterine fluids may be attributed to a significant presence of MPO precursor but also, to the presence of MPO inhibitors in these fluids. Further studies are needed to investigate the dynamics of MPO and its activity in the endometrium of mares in different reproductive situations.

Besides its balanced enzymatic functions, MPO has also been described to modulate neutrophil migration [78], increase TNF-α factor [79] and modulate cytokines [80]. These non-catalytic activities combined with our results during the physiological reproductive states of the uterus highlight the essential role of MPO in maintaining equine endometrial homeostasis even when not enzymatically active.

## 5. Conclusions

This is the first report characterizing MPO in the endometrium of healthy mares. MPO in the uterus of mares is constitutively present during the reproductive cycle and independent of inflammation, which indicates a physiological role of the enzyme in equine reproduction.

Although further studies are necessary to confirm our hypothesis, based on its hormone-dependent endometrial expression and its biological functions, we strongly suggest that equine endometrial cells are the main uterine source of MPO under physiological conditions, which as part of the uterine mucosal immune system in mares, contributes to the prevention of endometrial contamination and inflammation.

## Figures and Tables

**Figure 1 animals-13-00375-f001:**
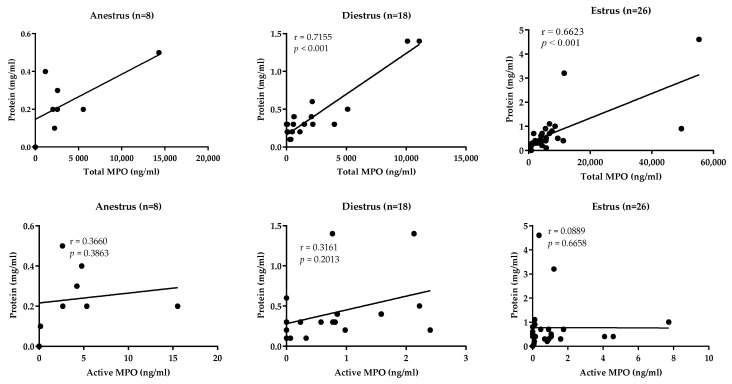
Association between total protein concentration and total and active myeloperoxidase (MPO) levels in uterine lavages in mares in the studied phases: anestrus, diestrus and estrus. *p* values < 0.05 are considered as significant.

**Figure 2 animals-13-00375-f002:**
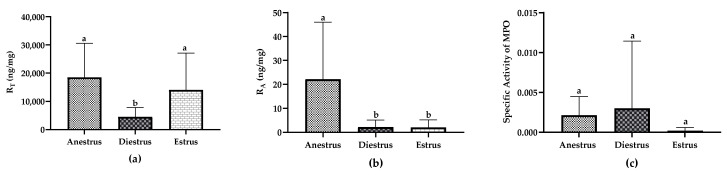
Myeloperoxidase (MPO) in uterine fluid of mares in anestrus, diestrus and estrus in absence of inflammation: (**a**) Total MPO/total protein ratio (R_T_), (**b**) Active MPO/total protein ratio (R_A_), (**c**) Specific Activity of MPO (active MPO/total MPO). Data are displayed as raw data (mean with SD). Columns with a different superscript are statistically different (*p* < 0.05).

**Figure 3 animals-13-00375-f003:**
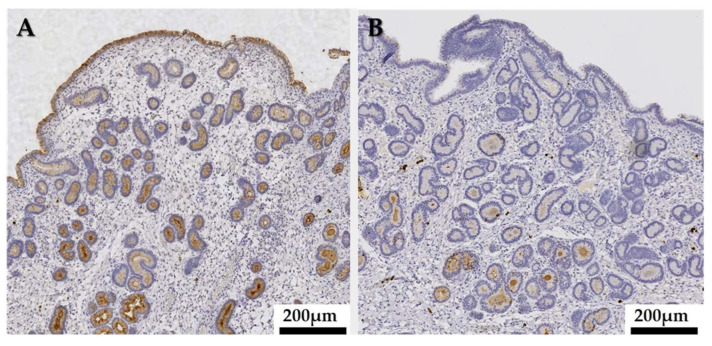
Immunohistochemistry. Endometrial myeloperoxidase (MPO) expression: comparison of MPO immunostaining of a mare through the same estrus cycle, (**A**) in estrus, a widespread distribution of immunopositive endometrial cells is observed; (**B**) In diestrus, endometrial cells showed only residual immunostaining. Magnification 10×.

**Figure 4 animals-13-00375-f004:**
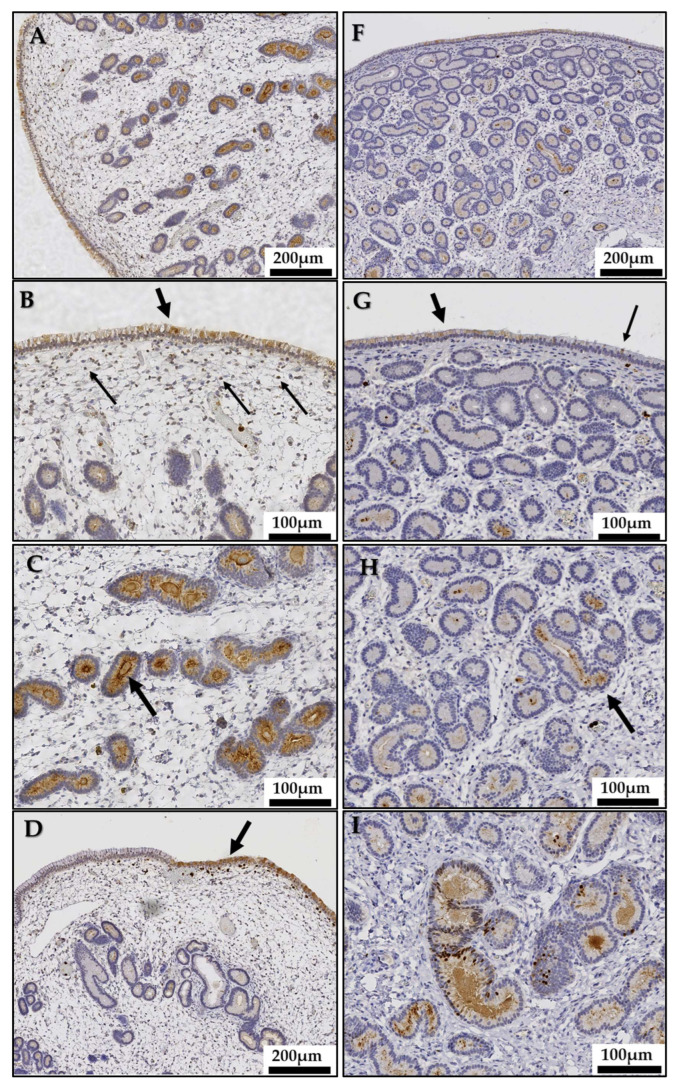
Immunohistochemistry. Equine endometrial myeloperoxidase (MPO) expression during the estrus cycle and seasonal anestrus. Images represent endometrial tissue samples from different mares in estrus (**A**–**D**), anestrus (**E**) and diestrus (**F**–**I**). Estrus: (**A**) Endometrial epithelial cells show a generalized MPO expression. The staining intensity in glandular cells increases with depth. Magnification 10×. (**B**) The adluminal epithelium shows a generalized intracytoplasmic diffuse mosaic-like staining reaction (thick arrow). A uniform intracytoplasmic pattern expression is observed in stromal cells (thin arrows). Magnification 20×. (**C**) Middle and basal uterine glands show a uniform, predominantly apical (arrow), intracytoplasmic diffuse pattern expression. Magnification 20×. (**D**) In presence of neutrophils in sub and transepithelial areas, the adluminal epithelium shows a uniform intracytoplasmic diffuse staining reaction (arrow). Magnification 20×. Anestrus: (**E**) Glandular endometrial cells are immunonegative. Magnification 10×. Inset: the luminal epithelium shows no or only a residual staining. Neutrophils in sub-epithelial capillaries are positively stained. Magnification 40×. Diestrus: (**F**) The epithelial cells immunostaining is not generalized. Magnification 10×. (**G**) The adluminal epithelium is irregularly marked. An intracytoplasmic diffuse mosaic-like staining reaction is observed in immunopositive areas (thick arrow) near immunonegative areas (thin arrow). Magnification 20×. (**H**) Endometrial glands show a sporadic MPO immunostaining (arrow). Magnification 20×. (**I**): Some basal glands show a nuclear immunostaining of individual cells sometimes accompanied by an intracytoplasmic diffuse staining. Magnification 20×. (**J**) Negative control showing no MPO in endometrial epithelial cells (left image, magnification 10×); and in neutrophils (right image, arrow, magnification 40×).

**Figure 5 animals-13-00375-f005:**
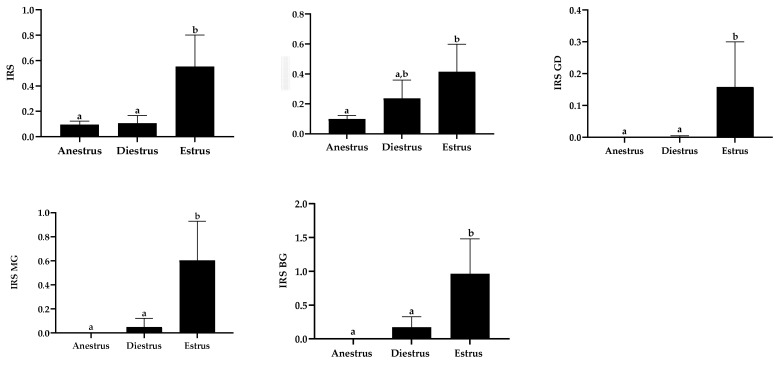
Myeloperoxidase immunoreactive score (IRS) among the studied phases in general and for each cell population. Data are displayed as raw data (mean with SD). Columns with a different superscript are statistically different (*p* < 0.05). LE: luminal epithelium; GD: glandular ducts; MG: mid glands; BG: basal glands.

**Figure 6 animals-13-00375-f006:**
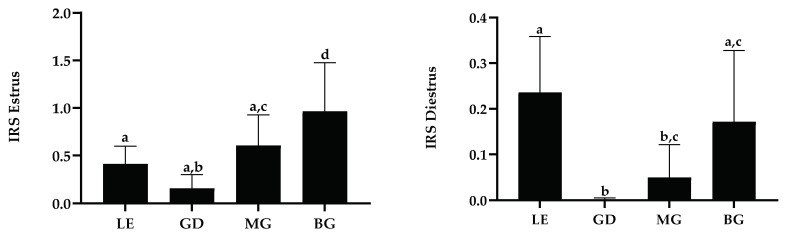
Myeloperoxidase immunoreactive score (IRS) in estrus and diestrus for each cell population. Data are displayed as raw data (mean with SD). Columns with a different superscript are statistically different (*p* < 0.05). LE: luminal epithelium; GD: glandular ducts; MG: mid glands; BG: basal glands.

**Table 1 animals-13-00375-t001:** Least squares means (+/− standard error) values of total and active myeloperoxidase (MPO), protein, total MPO/total protein ratio (R_T_) and active MPO/total protein ratio (R_A_) in uterine lavages in mares in the studied phases: anestrus, diestrus and estrus.

	Total MPO (ng/mL)	Active MPO (ng/mL)	Protein (mg/mL)	R_T_ (ng/mg)	R_A_ (ng/mg)
Anestrus	3798 (+/− 3464) ^a^	4.42 (+/− 0.84) ^a^	0.24 (+/− 0.26) ^a^	18512 (+/− 3994) ^a^	22.08 (+/− 3.46) ^a^
Diestrus	2319 (+/− 2334) ^a^	0.52 (+/− 0.54) ^b^	0.41 (+/− 0.17) ^a^	4320 (+/− 2517) ^b^	1.75 (+/− 2.04) ^b^
Estrus	8867 (+/− 1947) ^a^	0.85 (+/− 0.36) ^b^	0.77 (+/− 0.14) ^a^	13930 (+/− 2132) ^a^	1.48 (+/− 1.06) ^b^

Values with a different superscript within the same column are statistically different *p* < 0.05.

## Data Availability

Data is available under request to the authors.

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
