# Peer review of "Characterization of Myeloperoxidase in the Healthy Equine Endometrium"

_animals, 2023, doi:10.3390/ani13030375_

Round 1

Reviewer 1 Report

The study by Parilla Hernandez et al. proposed to characterize myeloperoxidase concentration in the equine endometrium throughout estrus cycle. This study seems like a follow-up of the study published back in 2014 by the same group, where only mares in estrus were analyzed. Also, this time not total myeloperoxidase, but also active myeloperoxidase was included in the analysis. The study design is interesting and results show possible implications for diagnosis of endometritis. However there are some aspects which need revision before acceptance for publication:

Lines 86-87: this sentences is a bit out of context here and repeats with other words the sentence in lines 82-83.

Lines 88-97: while first 2 paragraphs of the introduction are presenting lots of information in detail, these 3rd and 4th paragraph seem a bit "pushed" to just extend the introduction. The introduction is therefore not well balanced. In case no other information is available in the literature, maybe some information from the first 2 paragraphs can be moved to the 3rd and 4th paragraph to balance the introduction.

Lines 100-104: this information belongs rather to discussion in my opinion, this is the conclusion of the study

Material and methods, histology: one important aspect is missing (or not clearly stated), namely endometrosis. As authors themselves mention in the introduction, MPO is affected by presence of endometrosis. In this study, histology only analyzed the presence of neutrophils, but there can be endometrosis without inflammation as well. I doubt that all 36 mares included in this study had an endometrium biopsy score of I. This needs to be clarified and discussed in detail.

Material and methods, ELISA and SIEFED: dividing the samples in several subgroups to present the intra-assays CVs seems at least weird. I understand eliminating the samples below detection limit, but even for the other samples CVs are quite high, although another subset of samples is excluded from this quantification. It seems like the assays fail to detect MPO properly in a quite high percentage of samples, which questions the applicability of these methods.

Line 181: "non-specific" instead "non-specifics"

Line 190: "were" instead of "was"

Line 210: why least square means and not means and SD/SEM?

Lines 252-255: P value < 0.005 is awkward. was is really meant like this?

Figure 3: letters on the pictures should be top left (as in figure 4), not below

Figure 4: cannot be analyzed, as some pictures are overlapped

Discussion: was there any effect of age on MPO concentration? why was the MPO concentration not correlated to protein amount in the anestrus samples?

Author Response

Dear reviewer, 

Thank you for your remarks and comments. I think we have addressed them. Please find attached the point by point responses.

Thanks

Reviewer 2 Report

Generally, this is a solid, however descriptive, studies  characterizing presence, activity and localization of MPO in the uterus of mares under physiological conditions.  MPO and other neutrophils markers have been found before in mare’s uterus under pathological condition, ie. endometritis and endometrosis. The pathogenesis of endometritis and other disorders of uterine function in mares have recently been the subject of extensive research. Thus, the objective of this manuscript is very timely and important. This manuscript shown clearly for the first time that MPO is expressed in mare’s endometrial cells not only under pathological condition, but also during physiological estrous cycle. Intracytoplasmic immunoexpression of MPO was detected in the healthy endometrial epithelial cells, neutrophils and glandular secretions. These results confirm that constant presence of MPO in the uterine lumen of mares exists in absence of inflammation, probably as part of the uterine mucosal immune system, and suggest that endometrial cells are a source of uterine MPO under physiological cyclic conditions.  

The work is an extension of a previous studies by this and other research groups, in this meanings the originality and innovation of this paper is a little limited. Moreover, any physiological and functional data are done. However, the present data, may in concern with the pervious findings summarize the data how immune- as well other local, uterine factors  are regulated and may affect on reproductive processes in mares during the estrous cycle and early pregnancy under physiological and pathological conditions. The cellular and molecular mechanisms controlling the growth, structural changes and function of endometrium as well auto-, paracrine roles of PGs in domestic animals have recently been the subjects of extensive research. Moreover, the mechanisms involving in the regulation of the estrus cycle, luteolysis and opposite mechanisms - maternal recognition of pregnancy in domestic animals including mares are not well recognized.  Again, the objective of this manuscript is timely and important. Therefore, the manuscript include important data to contribute towards understanding the events of equine endometrium functions and should be published.

 The experiments appear to have been carried out carefully and the methods generally sound. A elegant and adequate experimental in vivo, ex vivo, as well analytical approaches were used.  In general the rationale, methodologies, and data are solid. I have only few remarks that may help to improve the manuscript.

 Specific points:

 Materials and Methods.

1.                  Too long and not well organized. A lot of detailed and repeating data and procedures is included. It is not needed to repeat and rewrite the methods and statements that were already published before by others and this research group. Only the original and new methods, not published before, should be detailed described. Please shorten this section.

2.                  The mares with various age have been used in the study. Therefore the carry should be taken on the clinical status of the animals, especially on the uterine disorders. Did authors check the patho-morphological status of endometrial to exclude endometrosis (periglandular fibrosis). Endometrosis is probably the most important reason for subfertility in older mares lowering  pregnancy rates. Moreover, endometrosis seems to affect particularly the maintenance of pregnancy causing both early embryonic losses and abortions affecting/modulating PGs production in the endometrium (See: Kenney 1978; Ferreira-Dias 1994; 1999; Hofmann et al. 2009; Szostek et al. 2012, etc.). Whether endometrial tissues were classified as category I according to the Kenney’s classification (Kenney RM. Cyclic and pathologic changes of the mare endometrium as detected by biopsy, with a note on early embrionic death. J. Am. Vet. Med. Ass. 1978; 72: 241-262 or Ferreira-Dias G, Nequin LG, King SS. Morphologic comparisons among categories I, II, and III equine endometrium using light or transmission electron microscopy. American Journal of Veterinary Research 1999; 60: 49-55).

This background procedure should be done before endometrial tissue using for experiments and analyzes. Some cellular and histological changes may occur in the endometrium without any clinical symptoms of endometrosis and/or endometritis.

3.                  Only immunolocalization and semi-quatitative analyzes of MPOs were done. How about mRNA and protein levels of examined enzymes in taken by biopsies endometrial tissues?  The authors should explain why the measure was done only on the immunolocalization level.

 Discussion and Conclusions.

Generally, the discussion needs to be streamlined and made more correlative than a rehash of the results. The discussion needs to be more about what this data could mean and why. As it stands, I think readers will get lost in the details and miss the points made in the discussion. The Discussion should explore the significance of the results of the work, not repeat them… Because no functional data showing the role of MPO in the uterus, the part of the discussion expelling role of MPO during the estrus cycle under physiological condition is rather speculative. Only selected mechanisms involved in the innate immune response was studies and preliminary data on production of MPO were done (no mRNA and protein expression, etc.). The methods used have some limitations and do not represent whole innate response, cell-to-cell contacts and interactions. Please see last paper of prof. G.M. Ferreia-Dias group (Amaral A, et al. Myeloperoxidase Inhibition Decreases the Expression of Collagen and Metallopeptidase in Mare Endometria under In Vitro Conditions. Animals (Basel). 2021 Jan 16;11(1):208.)

Author Response

Dear reviewer,

thanks for your comments and remarks. I hope we have addressed them and improved the manuscript. Please find attached the point by point responses to your remarks.

yours

Round 2

Reviewer 1 Report

Thank you for considering the comments and for improving the manuscript!

I consider that the information regarding exclusion of endometrosis is worth mentioning in the material and methods sections, otherwise other readers will critically ask themselves the same question. Also, for the histology pictures it is necessary to provide scale bars in order to comply with minimal information standards.

Author Response

Dear Reviewer,

as suggested we have added the exclusion of endometrial fibrosis in the material and methods section (L137-139) . We have also added the bars on the micrographs for ease of reading and left the magnification in the legend. Thanks again for your help in improving the manuscript.